# A gas breathing hydrogen/air biofuel cell comprising a redox polymer/hydrogenase-based bioanode

Julian Szczesny[1], Nikola Marković[1], Felipe Conzuelo [1], Sónia Zacarias[2], Inês A.C. Pereira [2], Wolfgang Lubitz[3], Nicolas Plumeré[4], Wolfgang Schuhmann [1] & Adrian Ruff [1]

Hydrogen is one of the most promising alternatives for fossil fuels. However, the power output of hydrogen/oxygen fuel cells is often restricted by mass transport limitations of the substrate. Here, we present a dual-gas breathing $H_2$/air biofuel cell that overcomes these limitations. The cell is equipped with a hydrogen-oxidizing redox polymer/hydrogenase gas-breathing bioanode and an oxygen-reducing bilirubin oxidase gas-breathing biocathode (operated in a direct electron transfer regime). The bioanode consists of a two layer system with a redox polymer-based adhesion layer and an active, redox polymer/hydrogenase top layer. The redox polymers protect the biocatalyst from high potentials and oxygen damage. The bioanodes show remarkable current densities of up to 8 mA cm$^{-2}$. A maximum power density of 3.6 mW cm$^{-2}$ at 0.7 V and an open circuit voltage of up to 1.13 V were achieved in biofuel cell tests, representing outstanding values for a device that is based on a redox polymer-based hydrogenase bioanode.

[1] Analytical Chemistry – Center for Electrochemical Sciences (CES), Ruhr-Universität Bochum, Universitätsstrasse 150, 44780 Bochum, Germany. [2] Instituto de Tecnologia Química e Biológica António Xavier, Universidade Nova de Lisboa, 2780-157 Oeiras, Portugal. [3] Max-Planck-Institut für Chemische Energiekonversion, Stiftstrasse 34–36, 45470 Mülheim an der Ruhr, Germany. [4] Center for Electrochemical Sciences (CES) – Molecular Nanostructures, Ruhr-Universität Bochum, Universitätsstrasse 150, 44780 Bochum, Germany. Correspondence and requests for materials should be addressed to W.S. (email: wolfgang.schuhmann@rub.de) or to A.R. (email: adrian.ruff@rub.de)

Molecular hydrogen generated from solar-driven water splitting in photoelectrochemical cells is promising for future energy technologies, as a sustainable and renewable alternative to fossil fuels[1–8]. The high amount of energy that is stored in the chemical bond of the $H_2$ molecule can be released in the form of electrons by using $H_2$-oxidation catalysts attached to an anode that is coupled to an $O_2$-reducing cathode in a $H_2/O_2$ fuel cell[6]. However, commonly used electrocatalysts for $H_2$ conversion are usually based on scarce and expensive materials containing noble metals[2]. An alternative approach includes the implementation of biocatalysts for the fabrication of $H_2/O_2$ biofuel cells (BFCs)[9,10]. In these biodevices, hydrogenases, with active centers based on earth-abundant metals (Ni and/or Fe)[11], have been proven to be powerful catalysts for the $H_2$ oxidation process at the bioanode with turnover rates similar to that reached with Pt[9,10,12–15]. Moreover, by employing $O_2$-reducing enzymes (e.g., multi-copper oxidases[16] such as bilirubin oxidase or laccase), remarkable power output of up to 1.7 mW cm$^{-2}$ [17] and open circuit voltage (OCV) of up to 1.17 V[18] have been achieved in $H_2/O_2$ BFCs with enzymes connected in a direct electron transfer (DET) regime to the electrode surface (for a recent overview on $H_2/O_2$ BFCs see refs.[9,10] and references cited therein). To further enhance the current densities and thus the power output of such BFCs, porous, high surface area[19] or gas diffusion electrodes[20,21] can be employed. The latter strategy circumvents limitations arising from slow mass transport and the low solubility of the gaseous substrate in the aqueous electrolyte by establishing a triple-phase boundary at the electrode/electrolyte/gas interface. This effect tremendously increases the local substrate gradient at the biocatalyst site by an enhanced substrate flux. Evidently, such gas diffusion layers are highly relevant for potential technological applications[20,21] and a theoretical power output calculated from the results obtained for the individual hydrogenase (bioanode) and bilirubin oxidase (biocathode) half cells of up to 8.4 mW cm$^{-2}$ has been reported[22].

Although hydrogenases reveal remarkable high turnover frequencies for the oxidation of $H_2$, their intrinsic instability against molecular $O_2$ and high potentials, which rapidly deactivate the enzyme, hampers their use in technologically relevant applications. In a DET configuration the enzyme may be directly exposed to detrimental oxygen traces and to high potentials during operational conditions in a $H_2/O_2$ fuel cell[23]. Consequently, the hydrogenase will be damaged under turnover conditions and hence suitable protection strategies, such as the previously proposed incorporation in a $O_2$-reducing viologen-modified polymer matrix[23,24], are required. Such low-potential polymer-based supporting matrices do not only eliminate harmful $O_2$ but also act as a Nernst buffer system and hence protect the sensitive catalyst from high-potential deactivation, which might occur in BFCs, especially if the anode is the limiting electrode[25,26]. Simultaneously, the redox polymer ensures faradaic communication between the biocatalyst and the electrode surface via a mediated electron transfer (MET) regime and allows for high biocatalyst loadings due to the 3D structure of the polymer matrix. Applying this strategy, it was possible to achieve outstanding $H_2$ oxidation currents for a flat polymer/hydrogenase electrode by incorporation of the highly active but sensitive [NiFeSe] hydrogenase from *Desulfovibrio vulgaris* Hildenborough (*Dv*H-[NiFeSe])[27] into a specifically designed viologen-modified polymer (poly(3-azido-propyl methacrylate-*co*-butyl acrylate-*co*-glycidyl methacrylate)-viologen designated as P(N$_3$MA-BA-GMA), Fig. 1)[28]. Moreover, $H_2/O_2$ BFCs comprising a polymer/hydrogenase bioanode with remarkable performances and stability became accessible[25,26]. However, the use of polymer/enzyme-modified gas diffusion bioelectrodes that provide not only a high-current density for $H_2$ oxidation but also are simultaneously able to protect the sensitive biocatalyst have not been reported yet, to the best of our knowledge. In addition, it can be expected that the porous structure of the gas diffusion electrodes does not only ensure a high substrate flux but also ensures short electron transport pathways within the polymer films, thus minimizing limitations arising from slow electron transport within the polymer matrix, which is still one of the major limitations when using redox polymers/enzyme films that are immobilized on flat electrodes[29,30]. Evidently, polymer/enzyme layers are promising systems for MET-type porous gas diffusion bioelectrodes, overcoming the limitations of deactivation of hydrogenases by $O_2$ and high potentials that arise when using a DET configuration[20], as well as slow electron transport in the polymer matrix and low catalyst loadings on flat electrodes[31].

Here, we present a polymer-based $H_2$/air gas-breathing BFC comprised of a viologen-modified polymer/hydrogenase bioanode and a bilirubin oxidase biocathode. The bioanode architecture ensures efficient protection against $O_2$ and high-potential deactivation even when the substrate $H_2$ is provided in gas-breathing mode and under anode-limiting conditions. Moreover, the proposed gas-breathing system reveals remarkable high-current density and power output that outperforms recently reported polymer/hydrogenase-based $H_2/O_2$ BFCs[25,26].

## Results

**Bioanode design.** To the best of our knowledge, all reports on high-current density hydrogenase gas-breathing electrodes rely on a DET regime[22,32,33], but such systems do not provide protection against $O_2$ and especially not against high-potential deactivation under bioanode-limiting conditions. Hence, the use of viologen-modified polymers acting as $O_2$-reducing matrix and simultaneously as Nernst buffer under turnover conditions is of high importance. For the preparation of a hydrogenase-based gas-breathing bioanode hydrophobic carbon cloth-based gas diffusion electrodes were first modified with a specifically designed more hydrophobic polymer adhesion layer, that is, the viologen-modified polymer P(GMA-BA-PEGMA)-vio (poly(glycidyl methacrylate-*co*-butyl acrylate-*co*-poly(ethylene glycol)methacrylate)-viologen, Fig. 1, blue, for synthesis and characterization see Methods section). A second layer consists of the P(N$_3$MA-BA-GMA)-vio/hydrogenase reaction layer (Fig. 1, pale red). The polymer P(GMA-BA-PEGMA)-vio which was used as adhesion layer reveals a lower viologen content (<65 mol%, see Methods) as compared to P(N$_3$MA-BA-GMA)-vio (71 mol%)[28] and hence exhibits a more hydrophobic character that facilitates the modification of the rather hydrophobic carbon cloth base material. Moreover, the underlying P(GMA-BA-PEGMA)-vio adhesion layer prevents an unwanted DET between the electrode and the redox protein.

The higher viologen content of P(N$_3$MA-BA-GMA)-vio ensures efficient protection of the hydrogenase within the reaction layer due to a fast electron transport by electron hopping. Despite the difference in viologen content, both polymers reveal identical redox potentials in 0.1 M phosphate buffer (PB) at pH 7 for the first viologen reduction process of $E_1 = -0.30$ V vs. SHE (see Supplementary Fig. 1d and ref.[33]), since both polymers were equipped with the same viologen unit. Hence, an electron transfer along both polymer layers to the electrode surface is possible. Moreover, the redox potential of the viologen units that is ≈150 mV more positive than the potential of the $H_2/2H^+$ couple at pH 7 (≈−0.45 V vs. SHE) should ensure a high driving force for the electron transfer between the polymer-bound mediator and the enzyme and is still low enough to ensure a significant OCV value for a corresponding BFC. Both [NiFe] hydrogenase from *Desulfovibrio vulgaris* Miyazaki F (*Dv*MF-

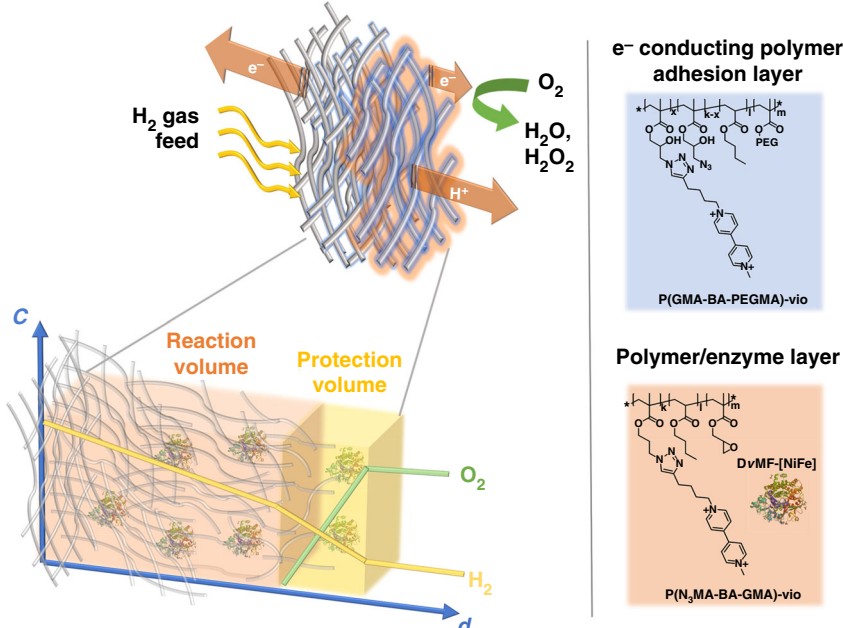

**Fig. 1** Schematic of the proposed gas-breathing polymer-based bioanode. The hydrogenase is electrically wired to the electrode surface via the viologen-modified redox polymers P(N₃MA-BA-GMA)-vio (poly(3-azido-propyl methacrylate-*co*-butyl acrylate-*co*-glycidyl methacrylate)-viologen) and P(GMA-BA-PEGMA)-vio (poly(glycidyl methacrylate-*co*-butyl acrylate-*co*-poly(ethylene glycol) methacrylate)-viologen, blue). The latter acts additionally as adhesion layer for the P(N₃MA-BA-GMA)-vio/hydrogenase reaction layer (pale red) due to its more hydrophobic character that enhances interactions with the hydrophobic surface of the carbon cloth gas diffusion electrode and prevents direct contact of the biocatalyst with the electrode surface, hence protecting the enzyme from high-potential deactivation (in combination with P(N₃MA-BA-GMA)-vio) as well as unwanted contribution from DET. At the polymer/electrolyte interface incoming $O_2$ is scavenged by the reduced polymer-bound viologen moieties (protection volume, yellow, left bottom). Expected normalized concentration profiles for $H_2$ (yellow) and $O_2$ (green) under turnover conditions are shown at the bottom (left) of the scheme illustrating protection from $O_2$ (with $c$ = concentrations and $d$ = distance from electrode). $O_2$ is reduced at the outer polymer/enzyme layer and hence the reaction layer remains unaffected. The porous structure of the carbon cloth-based electrode ensures a high polymer/biocatalyst loading and minimizes limitations due to slow electron transfer by keeping the electron transfer pathways short (see main text for further details). Note that the reaction as well as the protection layer contains the redox polymer and the biocatalyst. Not drawn to scale. *Dv*MF-[NiFe] = [NiFe] hydrogenase from *Desulfovibrio vulgaris* Miyazaki F (4U9H)[41]. PEG poly(ethylene glycol)

[NiFe])[34] and *Dv*H-[NiFeSe] were used as biocatalysts and embedded within the P(N₃MA-BA-GMA)-vio polymer layer. The adhesion and the reaction layers were deposited by a standard drop cast process from aqueous solutions. The prepared polymer double layer electrodes will be denoted as P(GMA-BA-PEGMA)-vio//P(N₃MA-BA-GMA)-vio/hydrogenase in the following.

Figure 2 shows cyclic voltammograms of a P(GMA-BA-PEGMA)-vio//P(N₃MA-BA-GMA)-vio/*Dv*MF-[NiFe] electrode under argon (black solid line) and $H_2$ atmosphere (red solid line) in gas-diffusion mode. The sigmoidal shape of the *I–E* curve under turnover conditions (red solid line), which is overlapping with the redox potential of the polymer-bound viologen moieties, convincingly demonstrates that the biocatalyst is productively wired via a MET regime. This is a prerequisite for an effective protection since the electrons for the reduction of $O_2$ are delivered from the hydrogenase upon $H_2$ oxidation[26,29]. Electrodes using *Dv*H-[NiFeSe] hydrogenase show similar behavior (Supplementary Fig. 2a). The experiments clearly demonstrate that, as expected, the underlying P(GMA-BA-PEGMA)-vio adhesion layer is capable of transferring electrons from the P(N₃MA-BA-GMA)-vio/hydrogenase reaction layer to the electrode surface. We want to note that due to the high absolute currents measured with the carbon cloth-based electrodes, the corresponding voltammograms may be affected by *iR* drop effects, which is indicated by an enhanced peak potential separation of the viologen redox couple of 72 mV compared to the values obtained for small 3 mm glassy carbon electrodes (44 mV for P(GMA-BA-PEGMA)-vio, Supplementary Fig. 1d). This

may also explain the slight offset between the half wave potential of the catalytic wave and the redox potential of the viologen-modified polymer.

The absolute currents under turnover conditions measured with the *Dv*MF-[NiFe] and *Dv*H-[NiFeSe] bioanodes were between 300 to 500 µA for both hydrogenases. Since the diameter of the first polymer layer of the polymer/hydrogenase bioanode (limiting electrode) is approximately 4 mm, current densities for the bioanodes were about 2.4 to 4 mA cm⁻². To the best of our knowledge, the gas-breathing electrodes largely outperform the best performing conventional flat polymer/hydrogenase-based bioanodes reported so far[33]. However, we want to emphasize that the calculation of current densities for porous electrode materials is just a rough approximation since the geometrical surface area is always smaller than the entire accessible surface area within the 3D structure, taking into account also the surface areas of all pores (vide infra)[17]. Nevertheless, for comparative purposes, both the absolute currents and current densities will be given in the following discussion. In addition, the nominal hydrogenase loading is provided for comparison.

The use of gas-breathing electrodes indeed enhances the catalytic currents obtained under turnover conditions. Cyclic voltammograms measured in gas-breathing mode (Supplementary Fig. 2b, red line) show significantly higher steady-state currents for polymer/*Dv*MF-[NiFe] films as compared with voltammograms measured when the substrate is purged through the electrolyte (blue line). The higher currents in gas-breathing mode demonstrate that the polymer is permeable to $H_2$ gas and

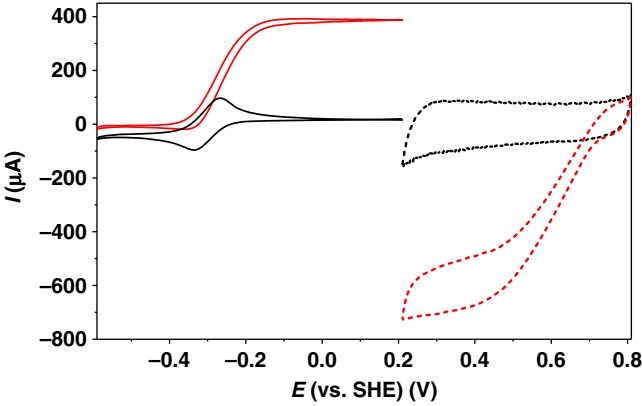

**Fig. 2** Electrochemical characterization of the bioelectrodes. Cyclic voltammograms of a P(GMA-BA-PEGMA)-vio//P(N$_3$MA-BA-GMA)-vio/ *Dv*MF-[NiFe] gas-diffusion bioanode (solid lines, left) and a *Mv*-BOx gas diffusion biocathode (dashed lines, right) in the absence (black lines) and presence of the respective substrate (red lines; bioanode: H$_2$; biocathode: air, no forced flow, passive breathing mode) in gas-breathing mode. Absolute currents are shown. Working electrolyte: 0.1 M phosphate buffer (pH 7.4); scan rates for all voltammograms: 10 mV s$^{-1}$; nominal hydrogenase loading (absolute value per bioanode/normalized by the area of the bioanode): 1.8 nmol/14.3 nmol cm$^{-2}$. P(N$_3$MA-BA-GMA)-vio poly(3-azido-propyl methacrylate-*co*-butyl acrylate-*co*-glycidyl methacrylate)-viologen, P(GMA-BA-PEGMA)-vio poly(glycidyl methacrylate-*co*-butyl acrylate-*co*-poly(ethylene glycol) methacrylate)-viologen, *Dv*MF-[NiFe] [NiFe] hydrogenase from *Desulfovibrio vulgaris* Miyazaki F; *Mv*-BOx bilirubin oxidase from *Myrothecium verrucaria*, SHE standard hydrogen electrode

that indeed higher local concentrations of the substrate are reached within the reaction layer even when a comparatively thick polymer matrix is used.

Moreover, the porous nature of the electrode seems to contribute to the pronounced catalytic currents obtained with the polymer/hydrogenase films: even when the substrate is provided in solution (H$_2$ bubbling through the electrolyte), the observed current densities are significantly higher than those obtained for a flat electrode system (cf. Supplementary Fig 2b, $J \approx$ 1.1 mA cm$^{-2}$, with results reported in refs.[26], $J \approx 0.5$ mA cm$^{-2}$). The higher currents may be attributed to a higher catalyst loading and/or by reducing the limitations arising from electron transport within thick redox polymer layers since the porous structure of the electrode increases the apparent amount of effectively wired catalyst, as it was demonstrated recently for the integration of redox polymer/photosystem II films in hierarchical structured porous indium tin oxide electrodes[30].

**Protection**. Steady-state currents under turnover conditions indicate that high-potential deactivation is fully prevented by incorporation of the hydrogenase into the redox polymer matrix. Protection from O$_2$ was tested in gas-breathing mode (100% H$_2$) and by bubbling a mixture of argon and O$_2$ (95%:5%) through the electrolyte. A current drop was observed when the polymer/enzyme films were exposed to O$_2$ (Fig. 3a). This observation is in analogy to the behavior that was observed for polymer/hydrogenase films deposited onto flat electrodes[25,26,33]. The decrease in oxidation current is caused by the consumption of electrons by the O$_2$ reduction reaction that takes place at the reduced viologen moieties[26,29]. After switching off the O$_2$ gas feed, the current is almost completely restored. The slight decrease of the absolute current (89% of the initial value at $t = 4500$ s) over several Ar/H$_2$/ O$_2$ cycles indicates that a small fraction of the biocatalyst was deactivated under anaerobic conditions. This fraction of the catalyst may be directly connected to the electrode surface in a

DET regime or may be embedded in rather thin parts of the polymer/enzyme films in which protection against O$_2$ is insufficient[29,33]. Both scenarios cannot be fully excluded since the formation of homogeneous films with constant thickness on the rather rough and porous electrode material is impeded. Nevertheless, under aerobic conditions a significant amount of the enzyme is effectively protected by the polymer matrix and the lifetime of the air-sensitive biocatalyst is largely extended as compared to electrodes operated in DET mode[26].

When O$_2$ was offered in gas-breathing mode, the O$_2$ concentration in the active layer seems to exceed the protection capability of the viologen matrix in this electrode configuration and the catalyst is irreversibly damaged (indicated by the only partial recovery of the oxidation currents when the O$_2$ gas feed was stopped, Supplementary Fig. 3a). In contrast, when the entire electrode was immersed into the electrolyte (non-gas-breathing conditions, low concentration of O$_2$ also from the back, Supplementary Fig. 3b), a reversible behavior was observed similar to the results depicted in Fig. 3a, and similar to the results reported in refs[25,26,33]. The results demonstrate that the proposed polymer/hydrogenase bioanode is highly suitable for the application in a H$_2$/O$_2$ BFC since protection from high potentials and from O$_2$ are guaranteed even for porous electrodes and in gas-breathing mode.

**Biofuel cell**. The polymer/hydrogenase bioanodes were coupled to an O$_2$-reducing gas-breathing biocathode that was equipped with a multi-copper oxidase, that is, bilirubin oxidase from *Myrothecium verrucaria* (*Mv*-BOx). The microporous, hydrophobic carbon cloth that was used was first chemically modified with a 2-aminobenzoic acid or 4-aminobenzoic acid (2-ABA or 4-ABA) layer in an electrochemical grafting process (see Methods for details) to increase the surface hydrophilicity of the material and to ensure a pronounced interaction with the O$_2$-reducing biocatalyst *Mv*-BOx due to a productive orientation of the enzyme on the electrode surface[22]. The enzyme was then immobilized on the chemically modified surface by means of a drop cast process. The potential for O$_2$ reduction of such electrodes under air-breathing conditions was +0.74 V vs. SHE (Fig. 2, dashed lines) and absolute currents of almost 800 µA were reached, thus ensuring limiting conditions of the bioanode.

The fully assembled membrane-free BFC containing the *Dv*MF-[NiFe] as H$_2$-oxidation catalyst at the bioanode exhibits a maximum power output $P_{max}$ of (259 ± 18) µW (Fig. 3b) or (2.05 ± 0.14) mW cm$^{-2}$ (Supplementary Fig. 4) at 0.7 V (average values over three independent experiments), respectively, with a nominal hydrogenase loading of 1.8 nmol per bioanode equivalent to 14.3 nmol cm$^{-2}$. The device reveals an OCV of (1.13 ± 0.03) V, which is similar to DET-based H$_2$/O$_2$ BFCs comprised of an air-sensitive hydrogenase and an O$_2$-reducing bilirubin oxidase (1.12–1.14 V)[22,31] and is even higher than values reported for systems equipped with O$_2$-tolerant hydrogenases (1.02 V)[17]. The low redox potential of the viologen-modified redox polymers ensures the envisaged high OCV value. However, this value exceeds the theoretical value of ≈1.04 V calculated from the onset potential of O$_2$ reduction at the biocathode (+0.74 V vs. SHE) and the redox potential of the viologen-modified redox polymers (−0.3 V vs. SHE) and may be affected by capacitive effects, contribution from DET and/or local pH gradients in the catalytic layer or at the electrode/electrolyte interface. The effect of an increase in OCV is further corroborated by the tailing that is observed in the power curves at the high voltage side, most likely due to a shift of the Nernst potential of the redox polymer that is induced by the enzymatic reaction which is continuously reducing the polymer-bound mediator species even at OCV.

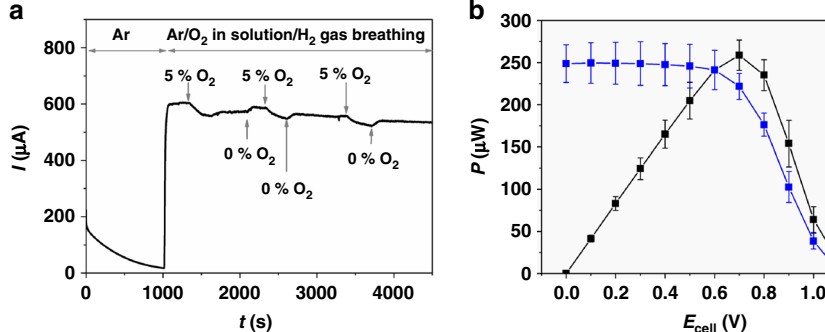

**Fig. 3** Protection and performance of the bioanode. Protection of *Dv*MF-[NiFe] from $O_2$ at the bioanode (**a**) and power output of the *Dv*MF-[NiFe]/*Mv*-BOD-based biofuel cell (**b**) in 0.1 M phosphate buffer (pH 7.4). **a**: Chronoamperometry of a P(GMA-BA-PEGMA)-vio//P(N₃MA-BA-GMA)-vio/*Dv*MF-[NiFe] bioanode at an applied potential of +0.16 V (vs. standard hydrogen electrode) and in $H_2$ breathing mode. The solution was purged with Ar (100%) or Ar/$O_2$ (95%/5%). **b** Power curve showing absolute power *P* and absolute current values *I* for a $H_2$/$O_2$ biofuel cell comprising a P(GMA-BA-PEGMA)-vio//P(N₃MA-BA-PEGMA)-vio/*Dv*MF-[NiFe] bioanode and the *Mv*-BOx biocathode measured with stepped potential chronoamperometry and in breathing mode. Average values for *P* and *I* calculated from three independent experiments are shown; error bars represent standard deviation. Source data are provided as a Source Data file. The maximum power output was determined to be 260 μW at 0.7 V. The cell reveals an open circuit voltage of (1.13 ± 0.03) V. Nominal hydrogenase loading (absolute value per bioanode/normalized by the area of bioanode): 1.8 nmol/14.3 nmol cm⁻². P(N₃MA-BA-GMA)-vio poly(3-azido-propyl methacrylate-*co*-butyl acrylate-*co*-glycidyl methacrylate)-viologen, P(GMA-BA-PEGMA)-vio poly(glycidyl methacrylate-*co*-butyl acrylate-*co*-poly(ethylene glycol) methacrylate)-viologen, *Dv*MF-[NiFe] [NiFe] hydrogenase from *Desulfovibrio vulgaris* Miyazaki F, *Mv*-BOx bilirubin oxidase from *Myrothecium verrucaria*

This mechanism was recently verified for a glucose/$O_2$-based BFC that contained a redox polymer-based bioanode[35] and is transposable to any fuel cell that contains a pseudocapacitive element, that is, a redox polymer matrix.

The *Dv*H-[NiFeSe]-based BFC (nominal biocatalyst loading: 1.53 nmol/12.1 nmol cm⁻²) shows a similar behavior (237 μW/1.88 mW cm⁻² at 0.7 V, OCV = 1.08 V, Supplementary Fig. 5). As indicated by the small standard deviations (Fig. 3b), the performance of various BFCs prepared with the same enzyme batch is highly reproducible. Moreover, cyclic voltammograms recorded with the polymer/hydrogenase bioanode before and after the BFC performance measurements (Supplementary Fig. 6a) show almost identical currents, indicating that the modified electrode is stable within the timescale of the BFC evaluation (≈1.3 h). During this time, a partial loss in activity was observed for the *Mv*-BOx biocathode (Supplementary Fig. 6b); however, the biocathode still provides a sufficient current to ensure anode-limiting conditions. To further increase the current output of the bioanodes, electrodes were modified with a higher amount of enzyme (by a factor of ≈2.2). Absolute currents of up to 1 mA for a *Dv*MF-[NiFe]-based bioanode (4 nmol per bioanode/31.8 nmol cm⁻², Fig. 4a) and 0.6 mA for a *Dv*H-[NiFeSe] (3.4 nmol/27.0 nmol cm⁻², Supplementary Fig. 7) were measured, which correspond to a current density of 7.9 and 5.3 mA cm⁻², respectively.

The power curve of a BFC containing the P(GMA-BA-PEGMA)-vio//P(N₃MA-BA-GMA)-vio/*Dv*MF-[NiFe] bioanode with high biocatalyst loading (31.8 nmol cm⁻²) shows a maximum power output of $P_{max} = 449$ μW at 0.7 V (Supplementary Fig. 8a) or 3.6 mW cm⁻² (Fig. 4b) if normalized to the surface area of the limiting bioanode. For these experiments, the biocathode was equipped with the double amount of *Mv*-BOx to ensure anode-limiting conditions.

It should be noted that the current and power output of the individual hydrogenase-based bioanodes (in the absence of $O_2$) and the BFC, respectively, clearly depends on the amount of loaded hydrogenase (Supplementary Fig. 9). On the other hand, a change in the pressure of the $H_2$ gas did not lead to increased current output, suggesting that indeed $H_2$ mass transport is not limiting in the gas-breathing mode.

Cyclic voltammograms recorded with the P(GMA-BA-PEGMA)-vio//P(N₃MA-BA-GMA)-vio/*Dv*MF-[NiFe] bioanode

and the *Mv*-BOx biocathode before and after BFC evaluation show no significant changes (Supplementary Fig. 8b, c). On the other hand, long-term measurements in BFC configuration at an applied cell voltage of 0.7 V show that after ≈7 h 50% of the initial current output were decayed under continuous operation in gas-breathing mode (Fig. 5, black trace). Within the first 2 h a rather stable current output was observed (Supplementary Fig. 10a). After ≈48 h the current levels to zero (Supplementary Fig. 10b, black line). Cyclic voltammograms recorded after the long-term measurement revealed a complete loss of activity for both electrodes (Supplementary Fig. 11). The decay of the current output of a single high-current density P(GMA-BA-PEGMA)-vio//P(N₃MA-BA-GMA)-vio/*Dv*MF-[NiFe] bioanode follows the same trend as that of the BFC (Supplementary Fig. 10b, red line), suggesting that the BFC is limited by the lifetime of the polymer/hydrogenase bioanode. Moreover, we conclude that deactivation of the hydrogenase by $O_2$ in the BFC system is not an issue, since the latter is absent in the half-cell experiment. Thus, the deactivation of the bioanode is most likely due to a slow desorption/disintegration of the polymer/enzyme layer triggered by the harsh conditions, that is, the extensive production of protons from $H_2$ oxidation within the active layer (local pH shift) at the carbon cloth surface during the highly efficient turnover, which may alter the surface properties of the electrode or disintegrate the polymer structure concomitantly decreasing attractive polymer–enzyme interactions.

**Polymer capping layer for enhanced stability**. To further enhance the stability of the active layer on the carbon cloth surface, we exploited the possibility of using a redox-silent but pH-sensitive polymer-stabilizing layer that was deposited on top of the active polymer/hydrogenase layer. For this, the redox-silent copolymer P(SS-GMA-BA) (poly(4-styrene sulfonate-*co*-glycidyl methacrylate-*co*-butyl acrylate), Supplementary Fig. 12)[36] was used, which bears acidic sulfonate groups that are deprotonated under neutral conditions and ensure a good solubility of the polymer in aqueous solution. Under acidic conditions, that is, under turnover conditions ($H_2$ oxidation, local increase of $H^+$ concentration), the sulfonate groups are protonated (Supplementary Fig. 12) and the polymer precipitates and prevents

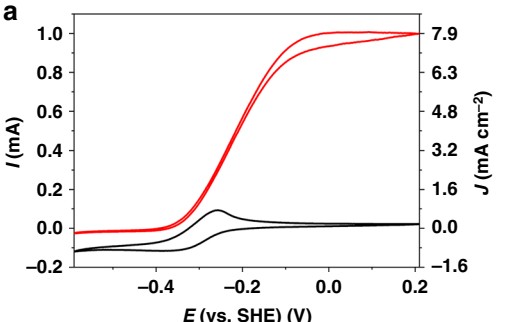
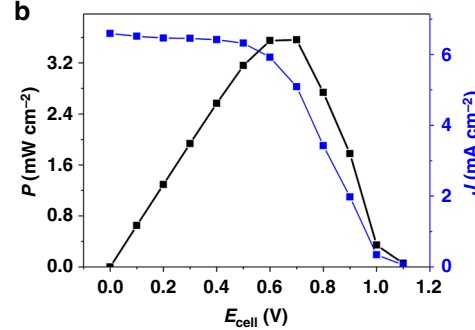

**Fig. 4** Characterization of the high-current density biofuel cell. Electrochemical characterization of the high-current density P(GMA-BA-PEGMA)-vio//P(N$_3$MA-BA-GMA)-vio/$Dv$MF-[NiFe]-based bioanode in 0.1 M phosphate buffer (pH 7.4) as half-cell (**a**) under argon (black line) and H$_2$ (red line) atmosphere (scan rate: 10 mV s$^{-1}$) and in the gas-breathing biofuel cell coupled to a $Mv$-BOx-based cathode (**b**). **a**, Left ordinate: absolute current $I$; right ordinate: current density $J$. **b** Power curve of the fully assembled one-compartment biofuel cell showing normalized power (left ordinate, black) and current values (right ordinate, blue). All measurements were conducted in gas-breathing mode. For the calculation of normalized current and power values, the spot size of the polymer layer was used, which was determined to be ≈4 mm. The biofuel cell reveals an open circuit voltage of 1.1 V and a maximum power density of 3.6 mW cm$^{-2}$ at 0.7 V. Nominal hydrogenase loading: 4 nmol/31.8 nmol cm$^{-2}$. P(N$_3$MA-BA-GMA)-vio poly(3-azido-propyl methacrylate-$co$-butyl acrylate-$co$-glycidyl methacrylate)-viologen; P(GMA-BA-PEGMA)-vio poly(glycidyl methacrylate-$co$-butyl acrylate-$co$-poly(ethylene glycol) methacrylate)-viologen, $Dv$MF-[NiFe] [NiFe] hydrogenase from $Desulfovibrio$ $vulgaris$ Miyazaki F, $Mv$-BOx bilirubin oxidase from $Myrothecium$ $verrucaria$, SHE standard hydrogen electrode

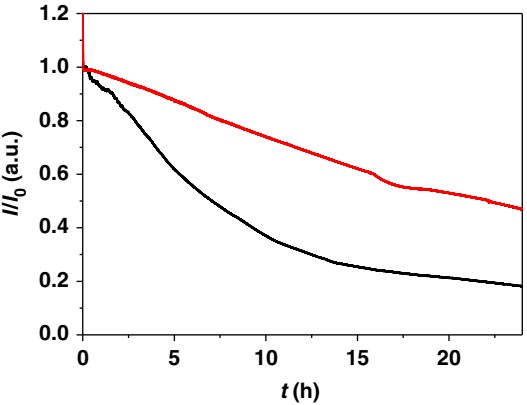

**Fig. 5** Long-term stability of the biofuel cell. Current output of the hydrogen/air-breathing biofuel cell comprising a P(GMA-BA-PEGMA)-vio//P(N$_3$MA-BA-GMA)-vio/$Dv$MF-[NiFe] bioanode (black trace) or a P(GMA-BA-PEGMA)-vio//P(N$_3$MA-BA-GMA)-vio/$Dv$MF-[NiFe] bioanode covered with a stabilizing P(SS-GMA-BA) layer (red trace) at a constant cell voltage of 0.7 V in 0.1 M phosphate buffer (pH 7.4) and the air-breathing $Mv$-BOx biocathode. Nominal hydrogenase loading: 4 nmol/31.8 nmol cm$^{-2}$. P(N$_3$MA-BA-GMA)-vio poly(3-azido-propyl methacrylate-$co$-butyl acrylate-$co$-glycidyl methacrylate)-viologen, P(GMA-BA-PEGMA)-vio poly(glycidyl methacrylate-$co$-butyl acrylate-$co$-poly(ethylene glycol) methacrylate)-viologen, $Dv$MF-[NiFe] [NiFe] hydrogenase from $Desulfovibrio$ $vulgaris$ Miyazaki F, P(SS-GMA-BA) poly(4-styrene sulfonate-$co$-glycidyl methacrylate-$co$-butyl acrylate). For comparison purposes normalized current values $I/I_0$ are shown, where $I_0$ is the initial current

desorption of the underlying active polymer/hydrogenase layer. Indeed, BFCs based on bioanodes modified with this pH-sensitive capping layer system show an enhanced stability: after 24 h the BFC still delivers ≈46% of its initial current (Fig. 5, red trace). Moreover, cyclic voltammograms recorded before and after the long-term BFC experiment (Supplementary Fig. 13) show that in terms of stability the biocathode is limiting: absolute currents of the biocathode measured under turnover conditions after the stability test show a stronger decrease than those of the bioanode. The isolated bioanode shows the same stability enhancement when covered with the P(SS-GMA-BA) capping layer (Supplementary Fig. 14).

## Discussion

We present a polymer-based hydrogenase bioanode that can be operated in gas-diffusion mode showing remarkable current densities of up to 8 mA cm$^{-2}$. The bioanode comprised a polymer adhesion layer that acts as a binder between the rather hydrophobic carbon cloth and the polymer/hydrogenase reaction layer. Voltammograms in gas-breathing mode show that higher currents are obtained in this configuration as compared to measurements in which the substrate was dissolved in the electrolyte solution. Hence, the used redox polymers are indeed permeable for the gaseous substrate. Moreover, also the 3D structure of the porous carbon cloth material contributes to the enhanced catalytic currents by increasing the amount of effectively wired biocatalyst. In contrast to hydrogenase connected via a DET regime, the proposed polymer-based electrode architecture provides an efficient protection of the sensitive biocatalyst against high potentials and against O$_2$. The fact that protection is also possible under gas-breathing conditions (H$_2$ is provided from the back side while the interfering O$_2$ is penetrating the polymer/enzyme film from the front) is consistent with proposed concentration gradients of the substrate H$_2$ (yellow trace) and the interference O$_2$ (green trace) depicted in Fig. 1 (bottom): a considerable amount of H$_2$ is penetrating the entire film and its concentration close to the polymer/electrolyte interface is sufficient for an effective protection. This is one of the main findings in this study and demonstrates that the benefits accompanied with the application of polymer/enzyme films, namely the protection against O$_2$ and high potentials, can be successfully combined with the advantages of gas-diffusion bioelectrodes, that is, the enhanced mass transport leading to high-current densities

The [NiFe]-based BFC delivers a remarkable maximum power output of 3.6 mW cm$^{-2}$ at 0.7 V. This value lies within the same order of magnitude than values obtained with the DET-based dual gas-breathing H$_2$/air (6.1 mW cm$^{-2}$ at 0.72 V)[22] and H$_2$/O$_2$ (8.4 mW cm$^{-2}$)[31] [NiFe]-hydrogenase/multi-copper oxidase biofuels cells reported by Kano and co-workers[18,20]. However, we want to emphasize that in the system reported in ref.[22] the BFC was operated under biocathode-limiting conditions (rather low potentials at the bioanode are ensured and O$_2$ is fully consumed at the biocathode). In the system reported in ref.[31], the power curve was not obtained experimentally, but only after mathematical estimation from the values obtained for the individual

half cells when $O_2$ is absent at the bioanode and the bioanode is not the limiting electrode. In contrast, the dual gas-breathing membrane-free $H_2$/air BFC described herein shows an unprecedented high-power output under bioanode-limiting conditions, largely outperforming our previously reported redox polymer/hydrogenase-based BFCs. At the same time, the developed redox polymer-based bioanode provides efficient protection against high-potential deactivation and $O_2$ even under the harsh conditions that were applied during BFC performance test. Moreover, the power density is significantly higher than that observed for hydrogenase/BOx $H_2$/$O_2$ fuel cells using rotating disk electrodes for enhanced mass transport (1.7 mW cm$^{-2}$ with an $O_2$-tolerant hydrogenase at 50 °C)[17]. The membrane-free and dual gas-breathing $Dv$H-[NiFeSe]-based BFC delivers a pronounced maximum power output of 237 μW/1.88 mW cm$^{-2}$ at 0.7 V, which is exceeding the maximum value of $P$ for the dual gas-breathing $H_2$/air BFC reported by Le Goff and co-workers[32] that was operated in a DET mode using the [NiFeSe] hydrogenase from *Desulfomicrobium baculatum* and $Mv$-BOx as biocatalysts (0.89 mW cm$^{-2}$ at 0.80 V)[32]. The here reported $Dv$MF-[NiFe]-based and the $Dv$H-[NiFeSe]-based systems show remarkable high OCV values of 1.13 and 1.08 V in the dual gas-breathing membrane-free one-compartment cell that is similar or even outperforms the previously reported polymer-based $H_2$/$O_2$ BFCs[25,26].

The stability of the bioanode could be largely enhanced by the introduction of a pH-sensitive polymer capping layer that was deposited on top of the active redox polymer/hydrogenase layer. In this multi-layer system, the desorption of the active layer is hampered and thus a constant power output in long-term experiments could be maintained. Since this capping layer is not directly interacting with the biocatalyst itself, it can be anticipated that this concept can be easily transposed to other fragile catalyst layers.

In conclusion, the proposed gas-diffusion redox polymer bioanodes introduce a new concept for the fabrication of highly efficient and protected bioanodes and will contribute to the successful integration of highly active but sensitive (bio)catalysts into technological and larger-scale applications.

## Methods

**Materials and chemicals**. All chemicals were purchased from Sigma-Aldrich, Merck, Acros Organics, abcr, Alfa Aesar, J.T. Baker, Fisher Chemicals, or VWR and were of reagent or higher grade. Deuterated solvents for nuclear magnetic resonance (NMR) spectroscopy were purchased from Deutero or Eurisotop and were stored at 4 °C. The copper(I) catalyst [Cu(MeCN)$_4$]PF$_6$ was stored in a desiccator at room temperature. The monomers glycidyl methacrylate (GMA) and butyl acrylate (BA) were passed through a short column containing the corresponding inhibitor remover (Sigma-Aldrich) and stored at −20 °C. The monomer poly(ethylene glycol) methacrylate (PEGMA) was first dissolved in isopropyl alcohol (50 mg mL$^{-1}$) and then passed through a column filled with inhibitor remover. The PEGMA/isopropyl alcohol solution was stored at room temperature. The synthesis of the polymer P(N$_3$GMA-BA-GMA)-vio and the alkyne-modified viologen unit vio was described earlier in ref[33]. The preparation and characterization of P(SS-GMA-BA) (stabilizing layer in the multi-layer configuration, nominal concentration of the co-monomers: SS = 50 mol%, GMA = 30 mol% and BA = 20 mol %, SS = 4-styrene sulfonate) was described recently in ref[36]. It was used as aqueous solution with a nominal concentration of 60 mg mL$^{-1}$. Spectroscopic and electrochemical data for all new compounds can be found in the synthesis part (vide infra) and in the Supplementary Information.

**Enzymes**. [NiFe] hydrogenase from $Dv$MF-[NiFe] was isolated and purified according to ref[37] and stored in MES buffer (pH 6.8) at −20 °C with a concentration of 200 μM. The isolation and purification of the recombinant form of the [NiFeSe] hydrogenase from $Dv$H-[NiFeSe] was described previously in ref[27]. The enzyme was stored in 20 mM Tris-HCl buffer (pH 7.6) with a concentration of 15 μg μL$^{-1}$ (170 μM) at −80 °C. Bilirubin oxidase from *Myrothecium verrucaria* (lyophilized powder, 15–65 U mg$^{-1}$ protein) was purchased from Sigma-Aldrich and stored at −20 °C.

**Spectroscopy**. All $^1$H-NMR spectra were recorded with a DPX200 or a DRX400 spectrometer from Bruker with a $^1$H resonance frequency of 200.13 and 400.13 MHz, respectively. The residual solvent peak was used as internal standard for calibration of chemical shifts. Ultraviolet–visible (UV–vis) measurements were performed with a Cary 60 spectrophotometer from Agilent Technologies in quartz cuvettes with an optical path length of 1 cm. Reflection Fourier transform infrared (FTIR) spectroscopy measurements were conducted on a Nicolet iS50 FTIR spectrometer or a Nicolet iN10 FTIR microscope from Thermo Fisher Scientific. Samples were prepared as thin films on corresponding substrates by a drop cast process from organic solvents (ethanol or acetone).

**Size-exclusion chromatography**. Size-exclusion chromatography (SEC) measurements for the determination of the molecular weight of the polymer backbone were performed with a PSS SECurity system against poly(styrene) standards with tetrahydrofuran (THF) as the eluent. The polymer sample concentration was 15 g L$^{-1}$.

**Electrochemical experiments**. All electrochemical experiments were conducted in a gas-tight sealed glass cell (Supplementary Fig. 15) under the respective atmosphere (argon, hydrogen, oxygen, or mixtures of those) at room temperature. Thermal mass flow controllers (GFC17, Aalborg Instruments and Controls) were used to adjust the desired gas atmosphere and gas mixtures with predefined compositions (for the exact composition of the gas feed see main text). Cyclic voltammetry and chronoamperometry measurements, as well as BFC characterization were performed with a Reference 600 (Gamry Instruments), a PalmSens2 v4.4 (PalmSens), or an Autolab FRA 2 Type III (Metrohm-Autolab) potentiostat. For experiments in three-electrode configuration (half-cell measurements) a Pt wire served as counter electrode and a Ag/AgCl/3 M KCl system served as reference electrode. For BFC measurements, the reference and counter electrode were both connected to the bilirubin oxidase biocathode. As working electrodes either glassy carbon electrodes with a diameter of 3 mm (characterization of the redox polymers) or round porous carbon cloth electrodes with a diameter of 2.1 cm (MTI, Carbon Foam Sheet, Porous C, 0.454 mm thick, ≈10 mL cm$^{-2}$ s$^{-1}$, porosity ≈31 μm, coated on one side with a Nafion/Teflon-based microporous film (50 μm), carbon content 5 mg cm$^{-2}$, EQ-bcgdl-1400S-LD) were employed (air-breathing and BFC measurements). Note that only one side of the electrode is coated with the Nafion/Teflon-based microporous layer. During electrode preparation only parts of the electrode surface were modified with the polymer/enzyme mixture by means of a drop cast process. To ensure that the bioanode is the limiting electrode, the modified surface area of the biocathode substantially exceeded the modified area of the bioanode in all experiments. Glassy carbon working electrodes were polished using alumina powder following standard protocols prior to the experiments. PB (0.1 M, pH 7.4) was used as electrolyte for all experiments. All potentials are rescaled with respect to the standard hydrogen electrode (SHE) according to $E_{SHE}$ = $E_{Ag/AgCl/3\,M\,\,KCl}$ + 210 mV. For BFC characterization, power curves were measured by stepped potential chronoamperometric experiments to minimize contributions from capacitive charging currents. After each potential step, steady-state currents were used to calculate the corresponding power values for the various BFCs. The back of the gas-breathing electrode was exposed to the corresponding gas atmosphere or to air. For polymer characterization, the corresponding polymer was drop cast onto glassy carbon electrodes and left to dry at room temperature. During the experiments in gas-breathing mode the electrochemical cell/electrolyte was continuously flushed with argon to prevent permeation of $O_2$ into the bulk solution.

**Bioanode preparation**. The porous carbon cloth electrodes (non-microporous side was used to avoid substantial DET in inhomogeneous polymer layers) were first modified by a drop cast process with P(GMA-BA-PEGMA)-vio that acts as an adhesion layer for the active P(N$_3$MA-BA-GMA)-vio/hydrogenase top layer. The latter was also prepared in a drop cast process. In brief, 20 μL of an aqueous solution of P(GMA-BA-PEGMA)-vio in water (7.5 mg mL$^{-1}$) were drop cast onto the carbon cloth-based material and dried at room temperature for at least 12 h (diameter of the polymer spot: ≈4 mm). P(N$_3$MA-BA-GMA)-vio (10 μL, 8 mg mL$^{-1}$) and the corresponding hydrogenase dissolved in buffer (3, 9, or 20 μL; $Dv$MF-[NiFe]: 200 μM; $Dv$H-[NiFeSe]: 170 μM) were premixed and then drop coated onto the adhesion layer and left to dry at 4 °C for at least 3 h. The so-prepared electrodes reveal nominal hydrogenase loadings of 0.6 nmol per electrode or 4.8 nmol cm$^{-2}$ (3 μL), 1.8 nmol/14.3 nmol cm$^{-2}$ (9 μL) or 4 nmol/31.8 nmol cm$^{-2}$ (20 μL) in case of the $Dv$MF-[NiFe] hydrogenase and 1.53 nmol/12.1 nmol cm$^{-2}$ (9 μL) or 3.4 nmol/27.0 nmol cm$^{-2}$ (20 μL) in case of $Dv$H-[NiFeSe], respectively. The nominal polymer loadings of P(GMA-BA-PEGMA)-vio and of P(N$_3$MA-BA-GMA)-vio were 150 μg/1.2 mg cm$^{-2}$ and 80 μg/0.6 mg cm$^{-2}$, respectively, for all bioanodes.

For bioanodes coated with the capping P(SS-GMA-BA) layer, the electrodes were first modified with the active polymer/hydrogenase layer as described above and then modified in a second drop cast process with a mixture of 20 μL of P(SS-GMA-BA) (60 mg mL$^{-1}$ in water) and 2 μL of the crosslinker 2,2′-(ethylenedioxy) bis(ethylamine) (1:37 vol% in water, the diamino-based crosslinker reacts with the epoxide units within the P(SS-GMA-BA) backbone and forms a stable 3D polymer crosslinker network). To ensure highest stability, the active layer was fully covered with the capping layer. The modified electrodes were then dried at 4 °C overnight.

**Biocathode preparation**. The biocathode were prepared by adapting a protocol that was described earlier in ref[38]. For the modification of microporous carbon cloth, the material was first hydrophilized with a 2-ABA or 4-ABA layer in an electrochemical grafting process (anodic oxidation of the amino group in 2-ABA or 4-ABA) in 0.1 M KCl/5 mM 2-ABA/water by applying a potential of +0.8 V vs. Ag/AgCl/3 M KCl for 60 s according to protocols described earlier in ref[22]. For the grafting process, the carbon cloth was pre-wetted with ethanol and rinsed with water and used immediately without drying. After grafting and after rinsing the electrode with water, 40 µL of a $Mv$-BOx solution (15 mg mL$^{-1}$ in 0.1 M PB, pH 7) was drop cast onto the ABA-modified microporous side of the carbon cloth electrodes (modification of the microporous Nafion/Teflon layer ensures a high loading of the $O_2$-reducing biocatalyst and thus anode-limiting conditions) and allowed to dry for 1.5 h at room temperature (nominal enzyme loading per electrode: 600 µg). For BFC experiments with the high-current density $Dv$MF-[NiFe] bioanodes (31.8 nmol cm$^{-2}$ hydrogenase loading), 80 µL of the $Mv$-BOx solution were used for modification (nominal enzyme loading per electrode: 1200 µg). All measurements were conducted in air-breathing mode under passive breathing (no forced air flow).

**Multistep synthesis of redox polymer for the adhesion layer**. Synthesis of the polymer backbone: The synthesis of the P(GMA-BA-PEGMA) backbone (poly (glycidyl methacrylate-co-butyl acrylate-co-poly(ethylene glycol)methacrylate), Supplementary Fig. 16) was conducted according to protocols reported in ref[39].

The radical initiator 2,2'-azobis(2-methylpropionitrile (AIBN) was recrystallized from hot toluene or methanol and stored at −20 °C. In a typical procedure, 5 g of the crude brownish initiator were suspended in ≈90 mL of methanol. The solution was carefully heated and then filtered to remove insoluble residues. Afterwards, the solution was cooled to 4 °C and kept at this temperature for 3–4 h. Colorless needle-like crystals were formed, filtered off, and dried in vacuo. The methanol solution was concentrated by removing the solvent under reduced pressure. After cooling to 4 °C overnight, a second fraction of needle-like crystals was obtained, filtered off, and dried under reduced pressure. Yield: 3.38 g colorless needles.

To a solution of 8 mL of isopropyl alcohol containing 0.4 g (0.8 mmol) of the monomer poly(ethylene glycol) methacrylate (PEGMA), the co-monomers GMA (0.711 g, 5 mmol) and BA (0.545 g, 4.3 mmol) were added under argon atmosphere. The reaction mixture was deaerated by argon bubbling and 5 mg of the purified radical initiator AIBN was added. The mixture was heated to 80 °C within ≈20 min and kept at this temperature for another 30 min until the solution became turbid. The polymer was precipitated by adding water. The colorless crude product was separated by centrifugation (4000 rpm, 30 min), suspended in 20 mL of MeOH, and precipitated again by adding 20 mL of water. The polymer was separated again by centrifugation (4000 rpm, 20 min), washed with 3 × 50 mL of diethyl ether, and dried in vacuo. The pure product was finally dissolved in 10 mL of dry dimethylformamide to yield a colorless polymer solution with a concentration of 60.3 mg mL$^{-1}$ (stored in the dark under argon). Yield: 0.603 g (38%). $^1$H-NMR (200.13 MHz, acetone-$d_6$) δ/ppm (Supplementary Fig. 17): 4.40/4.34 and 3.83 (all -CH$_2$-OCO- of GMA); 4.03 (-CH$_2$-OCO- of BA); 3.60 (-O-CH$_2$CH$_2$-O- of PEGMA); 3.26, 2.83, and 2.69 (epoxide moiety in GMA), 1.64 and 1.41 (-CH$_2$- of BA); 1.12 and 0.97 (overlapping, -CH$_3$ of GMA, BA, and PEGMA). The data are consistent with literature values[39]. Composition determined via integral ratio: GMA = 65 mol%, BA = 32 mol% and PEGMA = 3 mol% (nominal composition: GMA = 49.5 mol%, BA = 42.5 mol%, PEGMA = 8 mol%). SEC (THF, against poly (styrene) standard): $M_n$ = 34 kDa, PDI = 2.6. Reflection FTIR ṽ cm$^{-1}$ (Supplementary Fig. 18a): ≈2900 (s, multiple signals, C-H); 1721 (vs C = O).

Synthesis of the azide-modified polymer backbone: The synthesis of the azide-modified polymer (P(GMA-BA-PEGMA)-N$_3$, Supplementary Fig. 19) was conducted according to protocols described in ref[40]. To a solution of 10 mL dimethylformamide containing P(GMA-BA-PEGMA) in a concentration of 60.3 mg mL$^{-1}$ (total polymer mass = 603 mg with 62 wt% of GMA; this corresponds to 374 mg GMA and thus 2.6 mmol epoxide units) first 0.32 g (6 mmol) NH$_4$Cl and then 0.39 g (6 mmol) NaN$_3$ were added under an argon atmosphere. The slurry was heated to 50 °C and stirred for 22 h. After cooling down to room temperature, the polymer was precipitated by adding ≈90 mL of water. The colorless precipitate was separated by centrifugation (4000 rpm, 20 min) and washed with water (2 × 90 mL, with centrifugation after each washing step) and with ≈120 mL of diethyl ether. The latter was decanted off and the wet residue was suspended in 70 mL of methanol. The product was precipitated by adding 30 mL of water and separated again by centrifugation (4000 rpm, 30 min). Finally, the residue was washed with 40 mL of diethyl ether and dried under reduced pressure to yield 222 mg of a colorless product. The polymer was dissolved in 7 mL of acetone (31.7 mg mL$^{-1}$) and stored at 4 °C in the dark. $^1$H-NMR (400.13 MHz, acetone-$d_6$) δ/ppm (Supplementary Fig. 20): 4.66, 4.12, 4.01, 3.75 (-CH$_2$-OCO- moieties); 3.61 (-OCH$_2$CH$_2$O- of PEGMA); 3.41 (-CH$_2$-N$_3$), 2.84 (-OH), 1.93, 1.64, and 1.43 (-CH$_2$- moieties); 1.11, 1.05, and 0.97 (-CH$_3$ units). Composition: complete conversion of GMA units into -CHOH-CH$_2$-N$_3$ moieties, GMA signals are absent, N$_3$units = 65 mol%, BA = 32 mol%, PEGMA = 3 mol%. Reflection FTIR ṽ cm$^{-1}$ (Supplementary Fig. 18b): 3456 (s, broad, -OH); ≈2900 (s, multiple signals, C-H); 2104 (vs N$_3$); 1727 (vs C = O).

Synthesis of the viologen-modified polymer: The synthesis of the viologen-modified polymer P(GMA-BA-PEGMA)-vio (Supplementary Fig. 21) was conducted following procedures reported in ref[33]. Under an argon atmosphere, an acetone solution of the azide-modified polymer P(GMA-BA-PEGMA)-N$_3$ (2 mL, 34 mg mL$^{-1}$, total polymer mass = 68 mg with a GMA-N$_3$-unit ratio of 68 wt%; this corresponds to 46.24 mg N$_3$ units and thus 0.25 mmol N$_3$ functions) was diluted with 4 mL of acetone containing 110 mg (0.2 mmol, 0.8 eq.) 1-hex-5-ynyl-1'-methyl-4,4'-bipyridinium·2PF$_6$ (vio)[33]. Then, the copper(I) catalyst [Cu(MeCN)$_4$] PF$_6$ (74 mg, 0.2 mmol) followed by 1 mL of deaerated dimethyl sulfoxide (DMSO) was added while the reaction mixture was gently purged with an argon stream to remove remaining $O_2$. The homogeneous solution was heated to 50 °C and stirred overnight. The yellow reaction mixture was cooled down to room temperature and quenched with ≈6 mL of diethyl ether under stirring to induce the precipitation of the polymer. The organic phase was decanted off and the residue was successively washed with 3 × 10 mL of diethyl ether, 2 × 10 mL of water and with 3 × 10 mL of diethyl ether. Finally, the PF$_6^-$ salt of the product was dried under reduced pressure to obtain a pale green-yellow powder. Yield: 186 mg (crude product). $^1$H-NMR (400.13 MHz, DMSO-$d_6$) δ/ppm (Supplementary Fig. 22): 9.37 (s, -CH-aromatic viologen core); 9.26 (s, -CH- aromatic viologen core); ≈8.73 (multiple signals, overlapping, -CH- aromatic viologen core); 7.82 (s, -CH- triazole unit); 5.43 (s, -OH); 4.72 (s, -CH$_2$-N$^+$ = ), 4.44 (s, = N$^+$-CH$_3$); 4.07 and 3.87 (both s, -O-CH$_2$- moieties in the backbone); 3.49 (s, -OCH$_2$CH$_2$O- of PEGMA); 2.034, 1.66, 1.49, 1.28 (overlapping, broad, -CH$_2$- moieties from BA and linker chain of the viologen); 0.84 (s, -CH$_3$ polymer backbone). $^{31}$P{$^1$H}-NMR (161.98 MHz, DMSO-$d_6$) δ/ppm (Supplementary Fig. 23): 144.2 (septet, $J_{P,F}$ = 713 Hz). UV–vis (DMSO) λ nm$^{-1}$ (Supplementary Fig. 24): 265 nm (freely diffusing viologen: 265 nm[33]). Reflection FTIR (PF$_6^-$ salt was used) ṽ cm$^{-1}$ (Supplementary Fig. 18c): ≈3400 (broad, -OH); 3135 and 3071 (triazole ring); 2993 (s, multiple signals, C-H); 2104 (remaining N$_3$); 1732 (vs C = O); 1644 (vs -C = N, viologen core); 1450 (s), 1157 (s, broad), 862 (vs PF$_6^-$). Mid-point potential of the first, $E_1$, and second, $E_2$, reduction, determined by cyclic voltammetry in 0.1 M KCl/water with a drop cast film (from acetone) on glassy carbon electrode: $E_1$ = −0.27 and $E_2$ = −0.66 V vs. SHE (Supplementary Fig. 1a).

For the purification (removal of remaining copper catalyst and free viologen, as well as the exchange of hydrophobic PF$_6^-$ counter ions with hydrophilic Cl$^-$ anions) a metathesis reaction with the polymer and aqueous KCl solution was conducted. For this, small portions of the polymer were dissolved in aqueous KCl (0.1 or 1 M) overnight (small amounts of DMSO can be added to ensure that the polymer is completely dissolved) and dialyzed against KCl by using ultracentrifugation and size-selective membrane filters (Vivaspin 500, Sartorius) with a molecular weight cut-off of 5 kDa. To remove excess Cl$^-$ ions, the polymer was washed with copious amounts of water by using ultracentrifugation and membrane filters. The concentration of the purified polymer was adjusted to be 7.5 mg mL$^{-1}$ in water. The colorless polymer solution was stored at room temperature. Estimated mid-point potentials in 0.1 M PB, pH 7.4; drop cast film on glassy carbon: $E_1$ = −0.30 V and $E_2$ = −0.63 V vs. SHE (first reduction: Supplementary Fig. 1d).

## Data availability

The data that support the findings of this study are available from the corresponding author upon request. The source data underlying Fig. 3b are provided as a Source Data file.

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

## Acknowledgements

We thank Dr. Ines Ruff (Thermo Fisher Scientific, Dreieich) for reflectance FTIR measurements and Melinda Nolten (Ruhr-Universität Bochum) for help with the polymer syntheses. We are grateful to Nina Breuer and Patricia Malkowski (both Max-Planck Institute for Chemical Energy Conversion, Mülheim) for the preparation of the [NiFe] hydrogenase from *Dv*MF. The work was supported by the Deutsche Forschungsgemeinschaft in the framework of the Cluster of Excellence RESOLV (EXC 1069) and DIP (LU 315/17-1/2), by the DFG-ANR within the projects SHIELD PL746/2-1 and ANR-15-CE05-0020, and by the Fundação para a Ciência e Tecnologia (Portugal) (Grants UID/Multi/04551/2013, LISBOA-01-0145-FEDER-007660 (co-funded by FCT/MCTES and FEDER funds through COMPETE2020), PTDC/BBB-BEP/2885/2014 and PhD fellowship SFRH/BD/100314/2014. N.P. acknowledges funding by the European Research Council (ERC Starting Grant 715900).

## Author contributions

A.R., N.P., and W.S. conceived the study. A.R. prepared and characterized all polymers. S.Z. and I.A.C.P purified, characterized, and provided the NiFeSe hydrogenase. W.L. provided the NiFe hydrogenase. J.S. prepared the bioanodes and performed the electrochemical measurements. N.M. and F.C. prepared and characterized the biocathodes. A.R., J.S., N.P., and W.S. analyzed the results and interpreted the date. All authors contributed to the writing of the manuscript.

## Additional information

**Competing interests:** The authors declare no competing interests.

