## [Peer Review File · Nature Communications]

Reviewers' comments:

Reviewer #1 (Remarks to the Author):

This manuscript describes gas breathing H_2/O_2 biofuel cell using hydrogenase as an anodic catalyst and bilirubin oxidase as a cathodic catalyst, and porous carbon cloth was used as gas-diffusion electrode. The output performance of the biofuel cell was not bad, but very similar dual-gas-breathing H_2/O_2 biofuel cells have already been reported by Kano's group (Kyoto, Japan); the output performances were not so different. The electrode materials are different but essentially similar. The enzymes are not special. The viologen-based polymer for hydrogenase anode was also reported by the author's group in Nature Chem and other papers. Thus the reviewer feels that this manuscript is just a simple extension or combination of the past studies and I cannot find novelty for the publication in nature communications. I recommend to publish this in other specific journal, like J. Power Sources.

Reviewer #2 (Remarks to the Author):

The manuscript by J. Szczesny et al. describes electrochemical characterization of the dual-gas-breathing H_2 /air biofuel cell consisting of a hydrogenase based bioanode and a laccase based biocathode, where the bioanode was operated under bioanode limiting conditions. For the bioanode design the authors suggested a hydrophobic carbon cloth based gas diffusion electrode modified with two viologen-modified polymer layers: one is with more adhesive properties and the other is an active layer containing hydrogenase enzymes. To prepare the active layer the authors used [NiFe]- and [NiFeSe] hydrogenases from *Desulfovibrio vulgaris* Miyazaki F deposited on the viologen-modified polymer by a drop cast process. Describing electrochemical properties of bioanode, the authors demonstrated that the absolute currents under turnover conditions measured for both hydrogenases electrodes in gas breathing mode were 0.3-0.5 mA indicating that the biocatalyst is productively wired to the electrode surface via mediated electron transfer. Increasing in the amount of the enzyme applied on the bioanode results in the increase of the absolute current up to 0.6-1 mA. The authors show that in the gas breathing mode the catalytic currents for the bioanode were higher those currents obtained for measurements when H_2 (100 %) was purged through the solution. The authors tested the stability of the designed bioanode to oxygen in both a) gas breathing mode (using 95% of hydrogen and 5% oxygen) and b) by bubbling a mixture of 95% of argon and 5% of oxygen. The results show that in the gas breathing mode the bioelectrode is more sensitive to oxygen since the concentration of oxygen exceeds the protection capability of the viologen matrix. However, when the bioelectrodes were completely immersed into the electrolyte and purged with the gas mixture, they retain 89% of the initial activity after 75 min. Further the authors characterized fully assembled bifuel cell showing that the bioanode exhibits a maximum power output of 3.6 mW cm^{-2} at 0.7 V and was stable for ~1.3 h during the biofuel cell performance test. Long term measurements show that biofuel cell lost 50% of performance in 7 h, and after 48 h current fell to zero.

Overall the manuscript presents important findings on the design of the dual gas breathing H_2 /air biofuel cells, especially the implementation of two layer viologen-modified polymer electrodes which provide significant protection of the bioelectrode from oxygen and high potential deactivation. The authors' interpretations of the data are acceptable, but it is unclear if the work is of sufficient novelty and significance for publication in Nature Communications.

In spite of the new design of the two layer bioelectrode, the application of the gas breathing electrodes in fuel cells has been shown previously with maximum power density reached 6.1 mW cm^{-2} (at 0.72 V) (examples Gupta et al 2011, Lalaoui et al Chem Comm 2015, Kano et al 2016, Masurenko et al 2017 (review)). The application of the viologen-modified polymers (with different

chemical composition, than reported in the manuscript) as a protection of hydrogenases from high potential inactivation has been reported by Oughli et al (2018).

Some additional minor points that should be addressed:

1. Figure 4 should be modified, presenting high current density (panel A) and normalized absolute power (panel C), removing panels B and D. The panel B is showing the same results as C, and panel D is described in the text of the manuscript.
2. The authors should elaborate more on the stability of the bioelectrodes during long term measurements. Is it instability of the enzymes or simple desorption of the enzymes from the electrode surface result in the loss of the current during the biofuel cell performance?
3. How do different temperatures will affect the performance and stability of the described biofuel cell?

Reviewer #3 (Remarks to the Author):

The manuscript by Szczesny et al "A dual-gas 1-breathing H₂/air biofuel cell comprising a redox polymer/hydrogenase-based high current density bioanode" describes the development of a polymer-enzyme based H₂/O₂ fuel cell that addresses some limitations of previous work on similar enzyme based biofuel cell systems. The conductive polymer (N3GMA-BA-GMA) is combined with a redox dye (methyl viologen) for mediated transfer to the enzyme. The polymer functions in several ways; (i) diffusion layer for the concentration of H₂ delivery to H₂ase, (ii) protection of hydrogenase from O₂, (iii) rectifies the electrochemical potential ("Nernst buffer") to the potential of the redox couple of the dye, and 3D control of electron flux to wire enzymes to the flat 2D surface of the conducting electrode. The chemistry of the polymer, its integration and the biofuel cell design together represent the major contributions of this work. In this regard this cell addresses significant limitations of prior cell designs from the standpoint of O₂ and redox potential based inactivation of hydrogenase, as well as substrate limitation of gaseous substrates (H₂) due to poor mass transport and solubility in solution based cells. Otherwise the enzymes that are utilized as catalysts are basically the same as ones that have been used in prior designs and previously published works (and cited in the references).

As a result, this version of biofuel cell design enables a more effective coupling of the high catalytic rates of enzymes that sustain higher power densities and circuit voltages compared to prior work. While far below the best Pt-based fuel cell performance levels, the advantage of biofuel cells is the use of enzymes comprised of earth abundant materials. There is significant novelty on the chemistry and engineering principles that were used to create the cell, where mediated electron transfer is becoming a generalized strategy for controlling electron flux to and from enzymes, and whole cells, as a means for the conversion of chemical bond energy into electrochemical potential (and vice versa).

A weakness of the work is that there have been many publications on this concept and use of hydrogenases and oxygenases as catalytic materials for biofuel cells. In some respects, the work may be viewed as being incremental progress rather than a significant advancement. The fact that there is no validated or standardized testing regime to make comparisons of the operating values from this work to those published among the different reports in the literature, makes the authors' claim of higher performance difficult to fairly evaluate. This is more a criticism of the field, but should be taken into account and addressed by the authors in the write-up of the results and discussion section.

Regardless, the paper does report on a new design and engineering strategy that has significant merit for advancing the biofuel cell field. The work overall is well-formulated, and the experiments are well designed and address the most significant questions about the device performance

properties. I recommend publication once the following comments are addressed.

Comments:

- Page 7; lines 175-177 and page 8, lines 190-195. Did the authors test different levels of H₂ase loading on the bioanode current densities? Can the authors comment on what is the optimal loading value under 100% H₂? Is this value a limitation of overall device performance, or is the limitation mainly from O₂ inactivation of the H₂ases?
- Page 14, line 348 and lines 354-362. The term “benchmark” is somewhat misleading, as none of the work is truly benchmarked against a universal standard. The comparisons of power/current/voltage properties among different devices tested in different labs and operating conditions, not side-by-side under the same conditions, and thus should be framed more objectively as to reflect the lack of a standardized testing process.

Reviewer #1 (Remarks to the Author):

This manuscript describes gas breathing H_2/O_2 biofuel cell using hydrogenase as an anodic catalyst and bilirubin oxidase as a cathodic catalyst, and porous carbon cloth was used as gas-diffusion electrode. The output performance of the biofuel cell was not bad, but very similar dual-gas-breathing H_2/O_2 biofuel cells have already been reported by Kano's group (Kyoto, Japan); the output performances were not so different. The electrode materials are different but essentially similar. The enzymes are not special. The viologen-based polymer for hydrogenase anode was also reported by the author's group in Nature Chem and other papers. Thus the reviewer feels that this manuscript is just a simple extension or combination of the past studies and I cannot find novelty for the publication in nature communications. I recommend to publish this in other specific journal, like J. Power Sources.

We thank the reviewer for his/her evaluation of our manuscript. However, we disagree with the reviewer that this work is just a "simple extension" of our previous papers. The situation at a gas-breathing porous electrode is significantly different from the one found at a conventional flat electrode with substrate coming from the solution phase. To transpose the proprieties that go along with a viologen-modified polymer matrix, i.e. protection from high potential deactivation and O_2 protection, a careful polymer design and a wisely chosen electrode architecture must be applied. Since the transport pathways of the substrate and of detrimental O_2 are different in the gas-breathing system the transition from a conventional system (flat electrode, substrate/interference in solution) to the 3D structured gas breathing electrodes (porous structure, substrate from the back, interference from solution) is not straightforward and the question that was raised in the literature, *can redox-polymers act as enzyme immobilization and protection matrix in gas-breathing systems* (see Kano et al, Curr. Opin. Electrochem, 5, 173–182, 2017) showed that this system is of particular importance for the field of polymer based high performance biofuel cells and underlines the importance of our work.

Thus, we feel confident that our findings provide an essential proof-of-concept. Moreover, the high current output and the protection ability based on the polymer matrix that is achieved with the proposed system clearly demonstrates the benefits of the combination of a specifically designed redox polymer matrix and a gas-breathing system and outperforms all other conventional polymer-based hydrogenase biofuel cells that were reported previously. Moreover, we want to point out, as Reviewer 3 correctly mentioned, a direct comparison of the achieved power output with biofuel cells based on different approaches is not straightforward due to the different conditions, electrode materials, DET vs MET and different biocatalysts that are used. Thus, the here proposed biofuel cell should not only be judged in terms of performance and numbers but rather in terms of the novel concept that combines the advantages of a gas breathing system (high currents) and the redox polymer matrix (protection) which was presented for the first time.

Moreover, stimulated by Reviewer 2 we further investigated the stability of the bioanode. By this, we developed a general concept to enhance the stability of a fragile catalyst layer by introducing a functional stabilizing layer on-top of the active layer. By careful design of this polymer-based top-layer, i.e. making use of a pH sensitive polymer backbone, it was possible to drastically enhance the lifetime of the bioanode (see Reviewer 2). Since the polymer based stabilizing layer works independently from the catalyst, it can be anticipated that this protection concept can be transposed to any fragile catalyst layer and further adds novelty to our work.

Hence, we think that our work is not just an incremental step but adds valuable knowledge to the field and thus deserves indeed high attention.

Reviewer #2 (Remarks to the Author):

The manuscript by J. Szczesny et al. describes electrochemical characterization of the dual-gas-breathing H₂/air biofuel cell consisting of a hydrogenase based bioanode and a laccase based biocathode, where the bioanode was operated under bioanode limiting conditions. For the bioanode design the authors suggested a hydrophobic carbon cloth-based gas diffusion electrode modified with two viologen-modified polymer layers: one is with more adhesive properties and the other is an active layer containing hydrogenase enzymes. To prepare the active layer the authors used [NiFe]- and [NiFeSe] hydrogenases from *Desulfovibrio vulgaris* Miyazaki F deposited on the viologen-modified polymer by a drop cast process. Describing electrochemical properties of bioanode, the authors demonstrated that the absolute currents under turnover conditions measured for both hydrogenases electrodes in gas breathing mode were 0.3-0.5 mA indicating that the biocatalyst is productively wired to the electrode surface via mediated electron transfer. Increasing in the amount of the enzyme applied on the bioanode results in the increase of the absolute current up to 0.6-1 mA. The authors show that in the gas breathing mode the catalytic currents for the bioanode were higher those currents obtained for measurements when H₂ (100 %) was purged through the solution. The authors tested the stability of the designed bioanode to oxygen in both a) gas breathing mode (using 95% of hydrogen and 5% oxygen) and b) by bubbling a mixture of 95% of argon and 5% of oxygen. The results show that in the gas breathing mode the bioelectrode is more sensitive to oxygen since the concentration of oxygen exceeds the protection capability of the viologen matrix. However, when the bioelectrodes were completely immersed into the electrolyte and purged with the gas mixture, they retain 89% of the initial activity after 75 min. Further the authors characterized fully assembled biofuel cell showing that the bioanode exhibits a maximum power output of 3.6 mW cm⁻² at 0.7 V and was stable for ~1.3 h during the biofuel cell performance test. Long term measurements show that biofuel cell lost 50% of performance in 7 h, and after 48 h current fell to zero.

Overall the manuscript presents important findings on the design of the dual gas breathing H₂/air biofuel cells, especially the implementation of two-layer viologen-modified polymer electrodes which provide significant protection of the bioelectrode from oxygen and high potential deactivation. The authors' interpretations of the data are acceptable, but it is unclear if the work is of sufficient novelty and significance for publication in *Nature Communications*.

In spite of the new design of the two layer bioelectrode, the application of the gas breathing electrodes in fuel cells has been shown previously with maximum power density reached 6.1 mW cm⁻² (at 0.72 V) (examples Gupta et al 2011, Lalaoui et al Chem Comm 2015, Kano et al 2016, Masurenko et al 2017 (review)). The application of the viologen-modified polymers (with different chemical composition, than reported in the manuscript) as a protection of hydrogenases from high potential inactivation has been reported by Oughli et al 2018).

We thank the reviewer for his/her overall positive evaluation of our manuscript and that he/she feels that the work reveals important findings on the design of gas-breathing H₂/air fuel cells. Indeed, viologen modified redox polymers were used earlier for the protection of hydrogenases. However, in this work we show a completely new concept by transposing the viologen-protection properties to a gas-breathing system (ensuring high current densities and thus a prerequisite for potential technological applications) in which a completely new situation is established with respect to substrate/interference diffusion profiles (now H₂ from the back, O₂ from the front) and electrode architecture (moving from flat to porous 3D structured electrodes). A straightforward transformation from flat electrodes to the gas-breathing system cannot be expected, however, was solved by a careful polymer design and a well thought-through electrode modification process. Thus, we feel

confident that these findings will have a major contribution to the design of new polymer-based high-performance biofuel cells. See also Reviewer 1.

Moreover, in our revised version we further evaluated the stability of the bioanode and a new concept for enhancing the lifetime of the electrode by implementing a polymer multilayer system was introduced which is likely to be transposed to other catalytic systems with limited lifetimes (see item 2.).

Some additional minor points that should be addressed:

1. Figure 4 should be modified, presenting high current density (panel A) and normalized absolute power (panel C), removing panels B and D. The panel B is showing the same results as C, and panel D is described in the text of the manuscript.

Done. Panel B and D were moved to Supplementary Figures 8 and 9, respectively or to the new Figure 4.

2. The authors should elaborate more on the stability of the bioelectrodes during long term measurements. Is it instability of the enzymes or simple desorption of the enzymes from the electrode surface result in the loss of the current during the biofuel cell performance?

As evidenced by the CV measurements performed under turnover conditions after the long-term test (Supplementary Figure 10) both the bioanode and the biocathode are fully deactivated. Moreover, also signals of the polymer matrix at the bioanode are absent indicating that not only the enzyme but also the viologen modified polymer was disintegrated/desorbed during the measurements. Since the same trend is observed for the single bioanode (Supplementary Figure 10B red line) deactivation solely based on O_2 can be excluded (O_2 was absent in the half-cell measurement). Hence, the deactivation/disintegration is most likely based on the harsh conditions that are present under favoured turnover in the gas-breathing system. The extensive production of protons from H_2 oxidation may alter the surface properties of the carbon-cloth based electrode material or attacking the polymer/enzyme layer and thus reduce the attractive forces between the electrode material and the active layer.

Stimulated by the reviewers' comments, we exploited the possibility to use a second stabilizing layer that was deposited on top of the active polymer/hydrogenase layer to avoid desorption of the active layer during turnover. For this a redox-silent but pH-sensitive polymer matrix was used, that forms a stable film on the electrode surface under acidic conditions (precipitates upon protonation) and thus stabilizes the underlying active layer. The polymer multilayer system increased the stability of the system enormously: after 22 h still 46 % of the initial current remained in presence of the polymer-based stabilizing layer; as compared to a 50 % loss after 7 h for the single layer system. Since the stabilizing layer does not directly interact with the catalyst itself it can be anticipated that this concept can be easily transposed to other catalytic system with limited stability.

We expanded the discussion on the stability in the main text accordingly and we added a new chapter that discusses the two-layer system.

"The decay of the current output of a single high current density P(GMA-BA-PEGMA)-vio//P(N₃MA-BA-GMA)-vio/DvMF-[NiFe] bioanode follows the same trend than that of the BFC (Supplementary Figure 10b, red line) suggesting that the BFC is limited by the lifetime of the polymer/hydrogenase bioanode. Moreover, we conclude that deactivation of the hydrogenase by O_2 in the BFC system is not an issue, since the latter is absent in the half-cell experiment. Thus, the deactivation of the bioanode is most likely based on a slow desorption/disintegration of the polymer/enzyme layer triggered by the harsh conditioners, i.e. the extensive production of protons from H_2 oxidation within the active layer (local pH shift), at the carbon-cloth surface during the highly efficient turnover which

may alter the surface properties of the electrode or the disintegrate the polymer structure and thus reduce attractive polymer-enzyme interactions.

Polymer multilayer system for enhanced stability

To further enhance the stability of the active layer on the carbon cloth surface, we exploited the possibility of using a redox silent but pH sensitive polymer stabilizing layer that was deposited on top of the active polymer/hydrogenase layer. For this the known redox silent copolymer P(SS-GMA-BA) (poly(4-styrene sulfonate-co-glycidyl methacrylate-co-butyl acrylate), Supplementary Scheme 1)³⁶ was used that bears pH sensitive sulfonate groups that are deprotonated under neutral conditions and thus ensure a good solubility of the polymer matrix in aqueous solution. However, under acidic conditions, i.e. under turnover conditions (H_2 oxidation, local increase of H^+ concentration) the sulfonate groups are protonated and the polymer precipitates and should thus prevent a desorption of the underlying active polymer/hydrogenase layer. Indeed, biofuel cells based on bioanodes modified with a pH sensitive stabilizing layer system show an enhanced stability: after 24 h the biofuel still delivers $\approx 46\%$ of its initial current (Figure 4, red trace). Moreover, cyclic voltammograms recorded before and after the long-term biofuel cell experiment (Supplementary Figure 12) show that in terms of stability the biocathode is limiting: absolute currents of the biocathode measured under turnover conditions and after the stability test are lower than those obtained of the bioanode which contrasts the experiments conducted with only the active polymer/hydrogenase layer. The single bioanode shows the same stability enhancement when covered with the P(SS-GMA-BA) layer (Supplementary Figure 13)."

Discussion:

The stability of the bioanode could be largely enhanced by the introduction of a pH sensitive polymer layer that was deposited on top of the active polymer/hydrogenase layer. In the two-layer systems the desorption of the active layer is hampered and thus a constant power output in long term experiments could be maintained. Since this stabilizing layer is not directly interacting with the biocatalyst itself, it can be anticipated that this concept can be easily transposed to other fragile catalyst layers.

3. How do different temperatures will affect the performance and stability of the described biofuel cell?

It is common knowledge that elevated temperature favours desorption of a polymer from the electrode surface. Moreover, both hydrogenases are not thermophilic enzymes. Thus, it is evident that in our case enhanced temperatures will rather decrease stability than increasing it. Also, with the experimental setup (gas-breathing configuration) temperature control is difficult (note that due to the one-compartment cell the biocathode would also be affected by a temperature change)

We agree that for many hydrogenases, like the thermostable NiFe hydrogenase from *Aquifex aeolicus* that is used for the fabrication of BFC with remarkable performance and stability mainly by the Lojou group, higher temperature favours higher current outputs and good stability. However, here the used conditions are a compromise for all components included in the BFC system ensuring a reliable performance and durability.

Reviewer #3 (Remarks to the Author):

The manuscript by Szczesny et al “A dual-gas 1-breathing H₂/air biofuel cell comprising a redox polymer/hydrogenase-based high current density bioanode” describes the development of a polymer-enzyme based H₂/O₂ fuel cell that addresses some limitations of previous work on similar enzyme based biofuel cell systems. The conductive polymer (N3GMA-BA-GMA) is combined with a redox dye (methyl viologen) for mediated transfer to the enzyme. The polymer functions in several ways; (i) diffusion layer for the concentration of H₂ delivery to H₂ase, (ii) protection of hydrogenase from O₂, (iii) rectifies the electrochemical potential (“Nernst buffer”) to the potential of the redox couple of the dye, and 3D control of electron flux to wire enzymes to the flat 2D surface of the conducting electrode. The chemistry of the polymer, its integration and the biofuel cell design together represent the major contributions of this work. In this regard this cell addresses significant limitations of prior cell designs from the standpoint of O₂ and redox potential based inactivation of hydrogenase, as well as substrate limitation of gaseous substrates (H₂) due to poor mass transport and solubility in solution based cells. Otherwise the enzymes that are utilized as catalysts are basically the same as ones that have been used in prior designs and previously published works (and cited in the references).

As a result, this version of biofuel cell design enables a more effective coupling of the high catalytic rates of enzymes that sustain higher power densities and circuit voltages compared to prior work. While far below the best Pt-based fuel cell performance levels, the advantage of biofuels cells is the use of enzymes comprised of earth abundant materials. There is significant novelty on the chemistry and engineering principles that were used to create the cell, where mediated electron transfer is becoming a generalized strategy for controlling electron flux to and from enzymes, and whole cells, as a means for the conversion of chemical bond energy into electrochemical potential (and vice versa).

A weakness of the work is that there have been many publications on this concept and use of hydrogenases and oxygenases as catalytic materials for biofuel cells. In some respects, the work may be viewed as being incremental progress rather than a significant advancement. The fact that there is no validated or standardized testing regime to make comparisons of the operating values from this work to those published among the different reports in the literature, makes the authors’ claim of higher performance difficult to fairly evaluate. This is more a criticism of the field, but should be taken into account and addressed by the authors in the write-up of the results and discussion section.

Regardless, the paper does report on a new design and engineering strategy that has significant merit for advancing the biofuel cell field. The work overall is well-formulated, and the experiments are well designed and address the most significant questions about the device performance properties. I recommend publication once the following comments are addressed.

We thank the reviewer for the in-depth evaluation of our manuscript and we are happy that he/she finds that our manuscript presents a significant contribution to the design of novel biofuel cells. We hope that our revised version clarifies all remaining issues raised by the reviewer.

We agree with the reviewer that a comparison of biofuel cells based on different approaches (DET vs MET, gas breathing vs forced convection etc.) is indeed hampered or almost impossible. However, a comparison of previously reported polymer-based systems and the here proposed approach clearly demonstrates the benefits of the gas breathing architecture. Hence, with respect to the previously reported systems this biofuel cell shows indeed a high performance. To make this relation clearer we modified the text accordingly, see comment 2, below.

Moreover, going from conventional flat electrodes to a gas breathing system is not just an incremental step but rather requires a new and innovative approach in the design of the used polymer matrixes and the electrode architecture. We agree that the combination of both systems is obvious but still requires the adjustment of fundamental knowledge and the development of new designs to overcome the limitations that arise when using gas-breathing 3D structured electrodes, i.e. i) obstruction of the gas transport due to pore blocking by the polymer and ii) high potential deactivation in the 3D structured electrode (short distance between catalyst and electrode surface). (see also Reviewer 1 and 2 and point 2 below).

Comments:

Page 7; lines 175-177 and page 8, lines 190-195. Did the authors test different levels of H₂ase loading on the bioanode current densities? Can the authors comment on what is the optimal loading value under 100% H₂? Is this value a limitation of overall device performance, or is the limitation mainly from O₂ inactivation of the H₂ases?

We thank the reviewer for mentioning this very important point. Indeed, the performance of the biofuel cell depends on the amount of biocatalyst. As already shown in the manuscript, the power of the BFC is considerably increasing when increasing the enzyme loading from 14.3 nmol cm⁻² (Figure 3B, 260 μW at 0.7 V) to 31.8 nmol cm⁻² (Figure 4B, 449 μW at 0.7 V). Consequently, experiments with low catalyst loading (4.8 nmol cm⁻²) show a lower power output (137 μW at 0.7 V). The same trend was found for CVs measured with only the bioanodes. Thus, we conclude that contributions from O₂ deactivation in the BFC is not significant due to an efficient protection of the hydrogenase by the viologen modified polymer.

The fact that the currents and the power do not change when the pressure of the H₂ gas is changed indicate that H₂ mass transport is not limiting.

To make this even more clear we added a note on this to the main text and we added a new Figure to the Supplementary Information (Figure 9) that shows the plot of P and I vs catalyst loading:

It should be noted that the current and power output of the individual hydrogenase based bioanodes (in the absence of O₂) and the biofuel cell, respectively, depends on the amount of loaded hydrogenase (Supplementary Figure 9). A change in the pressure of the H₂ gas did not lead to increased current output, thus, we conclude that, as expected, H₂ mass transport is not limiting in gas-breathing mode.

Higher loadings, especially in case of NiFeSe, lead to unreproducible results most likely to loosely bound polymer-enzyme films that desorb quickly from the electrodes.

Page 14, line 348 and lines 354-362. The term “benchmark” is somewhat misleading, as none of the work is truly benchmarked against a universal standard. The comparisons of power/current/voltage properties among different devices tested in different labs and operating conditions, not side-by-side under the same conditions, and thus should be framed more objectively as to reflect the lack of a standardized testing process.

Indeed, the “benchmark discussion” in general can be rather misleading and overestimated. We agree with the referee that a direct comparison of different fuel cells is not straightforward, and care must be taken by analysing results reported in literature. However, what becomes obvious from our experiments is that the here proposed system provides the highest values ever reported for the previously reported viologen-modified polymer/H₂ase based bioanode/biofuel cells (See our previously published papers in Nature Chem 2014 and Angew. Chem 2015).

We changed the text accordingly to avoid the term “benchmark”:

Maximum power densities of 3.6 mW cm^{-2} at 0.7 V and an open circuit voltage of up to 1.13 V were achieved in biofuel cell tests **represent the highest values ever measured** for redox polymer-based hydrogenase bioanode.

And

Applying this strategy it was even possible **to achieve the highest values for H₂ oxidation currents reported so far** for a flat polymer/hydrogenase electrode by incorporation of the highly active but sensitive [NiFeSe] hydrogenase from *Desulfovibrio vulgaris* Hildenborough (DvH-[NiFeSe])²⁶ into a specifically designed viologen-modified polymer (P(N₃MA-BA-GMA)-vio, Figure 1).

And

Moreover, the proposed gas breathing system **reveals remarkable high** values for current densities and power output that outperforms recently reported conventional polymer/hydrogenase-based H₂/O₂ biofuel cells.^{24,25}

And

In contrast, the dual gas-breathing membrane free H₂/air biofuel cell described herein shows for the first time an unprecedented high-power output under bioanode limiting conditions, **and largely outperforms our previously reported redox-polymer/hydrogenase based biofuel cells.**

REVIEWERS' COMMENTS:

Reviewer #2 (Remarks to the Author):

After carefully reread the manuscript and reflecting on the response to my comments and those directed at the other reviewers my opinion is unchanged. The work is interesting and appears to be well performed but the study offers little in the way of basic science advancement and although demonstrates potential utility is in my opinion still and incremental advancement.

Reviewer #3 (Remarks to the Author):

The manuscript by Szczesny et al "A dual-gas 1-breathing H₂/air biofuel cell comprising a redox polymer/hydrogenase-based high current density bioanode" describes the development of a polymer-enzyme based H₂/O₂ fuel cell that addresses limitations of previous work on similar enzyme based biofuel cell systems by evolving a new electrode configuration for enzyme loading to enable the feed gases to flow through the enzyme layers. The work overall is well-formulated, my previous comments regarding the novelty, and evaluation of the work against prior studies have all been addressed. I recommend publication.

Response to Reviewers (in blue):

Reviewer #2 (Remarks to the Author):

After carefully reread the manuscript and reflecting on the response to my comments and those directed at the other reviewers my opinion is unchanged. The work is interesting and appears to be well performed but the study offers little in the way of basic science advancement and although demonstrates potential utility is in my opinion still and incremental advancement.

We thank the reviewer that he finds our work interesting and that it was well conducted. We believe that our findings are not only an incremental advancement but rather represent a fundamental step forward to stable and protected high current density H₂-oxidation bioanodes.

Reviewer #3 (Remarks to the Author):

The manuscript by Szczesny et al “A dual-gas 1-breathing H₂/air biofuel cell comprising a redox polymer/hydrogenase-based high current density bioanode” describes the development of a polymer-enzyme based H₂/O₂ fuel cell that addresses limitations of previous work on similar enzyme based biofuel cell systems by evolving a new electrode configuration for enzyme loading to enable the feed gases to flow through the enzyme layers. The work overall is well-formulated, my previous comments regarding the novelty, and evaluation of the work against prior studies have all been addressed. I recommend publication.

We thank the reviewer for his/her positive evaluation of our revised version and we are happy that he/she finds our manuscript suitable for publication.